# New Low Morphine Opium Poppy Genotype Obtained by TILLING Approach

**DOI:** 10.3390/plants12051077

**Published:** 2023-02-28

**Authors:** Jiří Červeň, Viktor Vrbovský, Jiří Horáček, Martin Bartas, Lenka Endlová, Petr Pečinka, Vladislav Čurn

**Affiliations:** 1Department of Biology and Ecology, Faculty of Science, University of Ostrava, Chittussiho 10, 710 00 Ostrava, Czech Republic; 2Research Institute of Oilseed Crops, Development and Research, Purkyňova 10, 764 01 Opava, Czech Republic; 3Agritec Plant Research, Ltd., Zemědělská 2520/16, 787 01 Šumperk, Czech Republic; 4Department of Genetics and Agricultural Biotechnology, Faculty of Agriculture, University of South Bohemia, Studentská 1668, 370 05 České Budějovice, Czech Republic

**Keywords:** opium poppy, TILLING, chemical mutagenesis, new breeding methods, expression profiles, morphine content

## Abstract

The opium poppy’s ability to produce various alkaloids is both useful and problematic. Breeding of new varieties with varying alkaloid content is therefore an important task. In this paper, the breeding technology of new low morphine poppy genotypes, based on a combination of a TILLING approach and single-molecule real-time NGS sequencing, is presented. Verification of the mutants in the TILLING population was obtained using RT-PCR and HPLC methods. Only three of the single-copy genes of the morphine pathway among the eleven genes were used for the identification of mutant genotypes. Point mutations were obtained only in one gene (*CNMT*) while an insertion was obtained in the other (*SalAT*). Only a few expected transition SNPs from G:C to A:T were obtained. In the low morphine mutant genotype, the production of morphine was decreased to 0.1% from 1.4% in the original variety. A comprehensive description of the breeding process, a basic characterization of the main alkaloid content, and a gene expression profile for the main alkaloid-producing genes is provided. Difficulties with the TILLING approach are also described and discussed.

## 1. Introduction

Opium poppy (*Papaver somniferum*) is one of the oldest cultivated medicinal and food plants that has been used since Neolithic times [1]. It can mostly be found as an important agricultural crop in subtropical and temperate climate regions, with the highest production in subtropical parts of the northern hemisphere [2].

The opium poppy is usually grown for medicinal products or for culinary reasons. This leads to two contradicting breeding strategies, that often follow opposing goals [3]. Varieties for medicinal and pharmacological purposes are usually selected for their high content of different benzylisoquinoline alkaloids. While more than 50 different alkaloids have been isolated from poppy plants, the main focus is on morphine, thebaine, papaverine, codeine, noscapine, and sanguinarine [4]. Even though these compounds have various uses in medicine, they can be relatively easily misused for the production of hallucinogenic compounds, of which heroin is the most common [5]. Although there have been many attempts to synthesize these compounds chemically, or to use cultured cells or genetically altered microorganisms, the complexity and chemical nature (including several optical isomers) hampers these endeavors, and production in plants still remains the most efficient way of their production [6]. When cultivated for culinary purposes, there are several criteria upon which the breeding is focused including: tons of seed per hectare; the composition of different oils, fatty acids and other beneficial nutrients; and taste [7]. Whilst poppy is grown for the production of seeds that do not naturally contain alkaloids, food-grade poppy varieties require that the alkaloid content in biomass be kept as low as possible due to the misuse of biomass and the contamination of seeds by alkaloids. Poppy cultivation is limited by different legislative regulations that vary in different countries. Some of the poppy varieties can be used for both alkaloid and food production, but the breeding goals tend to produce more specialized varieties. The most commonly found varieties for both applications are described in greater detail in a wide-ranging review by Labanca et al. [2]. Additional factors that are taken into account in the breeding process include resistance to various biotic and abiotic factors and sensitivity to pesticides/herbicides [8,9,10,11].

The opium poppy is usually diploid (2N) but some varieties or experimental breeding materials are tetraploid (4N) [2]. Moreover, about 8 million years ago, there was a whole genome duplication event [12]. In the evolutionary history of *Papaver somniferum*, there have also been some other genetic rearrangements which led to the whole or partial duplication of more genes, some of which are part of the benzylisoquinoline alkaloids metabolic pathways [13]. In the year 2018, the sequence of the whole opium poppy genome was published [14] and in 2019 this was updated for the benzylisoquinoline alkaloids’ production pathways [6]. Breeding techniques using colchicine-induced polyploidization for increasing alkaloid production led to interesting results, mostly in the work of Mishra et al. from 2010. In this work, differences have not been found in the whole alkaloid content, however, there has been a huge shift toward morphine content (from the original 25% to 40%) [15]. This can be partially explained by some recent findings of gene clustering in the benzylisoquinoline alkaloids’ production pathways [16].

Despite great developments in the field of molecular techniques in plant breeding, consisting of CRISPR/CAS9 with various modifications and other OMICs technologies [17], there are still legislative problems with GMOs mainly in Europe and USA. Moreover, the general public is often afraid of products that come from such genetic manipulation, and there still is some scientific discussion about GMO safety [18]. One of the ways of overcoming this problem is with “old school” chemical mutagenesis, often in conjunction with NGS or other “OMICs” tools [19,20,21,22,23]. Due to the difficult detection of mutations and the not entirely successful use of methods of induced mutagenesis in plant breeding, the approach of molecular or reverse genetics has been utilized, since 2000, by the TILLING method (Targeting Induced Local Lesions IN Genomes) [24]. TILLING is a method that allows for the controlled identification of mutations of a specific gene and combines a standard and effective technique of mutagenesis with a sensitive DNA screening technique to identify individual point mutations in the target gene [25,26]. This approach enables the efficient selection of new genotypes and their use in breeding [27].

The goal of this work was to introduce principles and methods of reverse genetics into the breeding process of the opium poppy. The aim was to obtain new varieties with a changed composition and total alkaloid content, especially varieties with a low morphine content, which would find use within the food industry. In this work, we combined traditional chemical mutagenesis using 1% ethyl methanesulfonate (EMS), with PACbio SEQUEL-SMRT sequencing of amplicons of genes of the metabolic pathway of alkaloid synthesis, which are found in the genome in only one copy for mutation screening. For the phenotypic selection and characterization of plants, HPLC of the main alkaloid content and gene expression of selected alkaloid-producing genes were used.

## 2. Results

### 2.1. Bioinformatical Selection of Marker Genes

First (spring, 2019), we checked the available genome information for partial or whole genome duplications affecting genes involved in the morphine production pathways. We focused on genes that were only found in one copy of the poppy genome and had similar whole gene lengths. The single gene copy was selected as if the mutation would be found in one locus; functional proteins could still be produced from other non-mutated copies of multicopy genes. Similar length is a criterion important for SEQUEL-SMRT sequencing, as there would be uneven coverage (larger coverage for shorter genes). Results are summarized in Table 1. We can see that some genes are found in multiple copies. Some genes are duplicated on the same chromosomes (*T6ODM*), some are found on different chromosomes (*NMCH*) and some exact positions were not yet known (marked as “Unplaced”). For some genes, we can see the whole gene duplication (*T6ODM*, *COR*), although for most of the genes, both CDS and whole gene lengths slightly differ. Based on the aforementioned criteria, we selected (S)-coclaurine-N-methyl transferase (*CNMT*), (S)-norcoclaurine synthase 2 (*NCS2*), and salutaridinol-7-O-acetyl transferase (*SalAT*) as selection marker genes.

### 2.2. Mutational Analysis of Selected Gene Markers

Detection of mutations was focused only on three genes from the morphine metabolic pathway. These three genes were found in only a single copy in the poppy genome. Genes that are duplicated were not used in this screening so that the discovered mutations were not masked by other (unmutated) gene copies. The morphine biosynthesis pathway can be seen in Figure 1.

Three selected genes were amplified by PCR to the maximum possible length. Amplicons were barcoded using SMRTbell 96 barcoded adapter plate 2.0 and sequenced using PacBio SEQUEL-SMRTII chemistry. Out of the 96 sequenced and possibly mutant plants, mutations were found in 6 of them. Mutation identification was carried out in the leaves of young plants, and six identified mutants of the M2 generation were further grown to produce seeds of the M3 generation. Mutations and their predicted effects are summarized in Table 2. Progeny were divided into individual cultivation plots named 9038–9070 (details in Table 2), so that in the next generation only plants that would carry the mutations after recombination events could be selected.

### 2.3. HPLC Analysis of Main Alkaloid Content in Selected M3 Plants

Offspring from selected genotypes were cultivated to full maturity and alkaloid content was measured by HPLC. As expected, we can see some variation even in genotypes coming from the same progenitor plants. However, in all the selected plants, there was a lower production of morphine (the average from all plants was 44%, and the minimum was 7% compared to the original genotype). Interestingly, in some plants there has been an increase in thebaine production, and plants from the least morphine-producing genotype (9042) had the highest thebaine content. In some genotypes, including the least morphine producing (9042), there was a drop in codeine production. In one of the genotypes, a significant increase in thebaine production was observed. All the results are summarized in Table 3. For the eight genotypes with the highest changes in alkaloid production (in general genotypes with the lowest morphine content), the gene expression of selected genes involved in the morphine production pathways was analyzed. These genotypes are marked as bold in Table 3.

### 2.4. Gene Expression Analysis of Selected Genotypes

For genotypes with the highest decrease in morphine or codeine production, gene expression analysis was performed for selected genes in the morphine production pathway. We have focused on genes in the main morphine production pathway, and have not analyzed genes leading to branches that produce other important alkaloids (e.g., papaverine, noscapine, and sanguinarine). Gene expression was monitored in three developmental stages of the poppy plant: bud initiation stage, flowering stage and capsule stage. Two types of samples were analyzed—thin slices across the whole sample, and only the outer epidermis with phloem. Expression profiles can be seen in Figure 2.

There is no clear pattern in the expression of analyzed genes, which can be partly explained by the fact that in the M3 generation there remains rather large heterogeneity among plants. Some genotypes show higher downregulation or upregulation of all pathways, however, this is not consistent through the developmental stages. In some samples, the expression of genes in the later part of the pathway is increased (e.g., Figure 2B, genotypes 9042, 9045, and 9063; Figure 2D, genotype 9046). The least morphine-producing genotype has the highest downregulation in all genes in the capsule stage, mainly in the epidermal/phloem sample (Figure 2F). However, there is no obvious correlation between the gene expression of all analyzed genes in all the developmental stages with the HPLC-assessed alkaloid content.

## 3. Discussion

A lack of suitable donors of traits and characteristics, narrow genetic variability and an unavailability of genetic resources are the key limits of classic breeding procedures. One of the possibilities for the creation of new genetic variability is induced mutagenesis and mutational breeding. However, this procedure, and especially obtaining the appropriate mutants, is very difficult and time-consuming. A new chance for improving this process is the TILLING technology [19,23]. TILLING, in combination with NGS, is thought to be a useful tool for the development of new plant varieties, however, it faces more difficulties. Generally, gene/genome duplication and/or polyploidy can make the analysis of mutations challenging [28]. To avoid these complications, we only focused on three genes found in only a single copy in the poppy genome that was available at that time. Selected genes (*CNMT*, *NCS*, and *SalAT*) are important in the morphine production pathway, and it has been shown that their inactivation would decrease morphine production [29]. Unfortunately, as we screened only for selected genes in the TILLING approach, we know nothing about mutations in other parts of the genome. This can be partly seen in our results, both from the alkaloid content and from the gene expression analysis. In the lowest morphine-producing genotype decrease both in codeine and morphine production was observed, but we can see an increase of thebaine content. This cannot be explained by a mutation in the *SalAT* gene that was used for screening. It is most likely that an inactivation of genes after thebaine production was induced. Interestingly, the *SalAT* gene catalyzes thebaine synthesis. It is possible that the mutation induced some genetic rearrangements, which in the end led to increased thebaine synthesis. This will be analyzed in later studies by the whole genome sequencing of this selected genotype. Notwithstanding these limitations, the TILLING approach is applicable for the creation and selection of genotypes with different patterns or alkaloid content and we have demonstrated the applicability of this approach in poppy breeding.

Studies of the expression of genes involved in morphine-producing pathways are more difficult, as expression occurs differently in different tissues. At least three types of cells are involved in the pathway [30], while different parts of the pathway take part in specific cells. However, the products of different steps (and maybe even mRNAs) are transferred among cells, as was recently described [31,32]. Moreover, it has been shown that the expression of particular genes tends to happen in clusters [16] however this was not clearly seen in our results. This can be partially explained by random mutations that might be present within our genotypes. More specialized techniques, for example, single-cell expression profiling, might help to better understand gene expression in TILLING-induced genotypes. In general, phenotypic traits such as alkaloid content seem to be a more useful selection criteria than gene expression profiles.

## 4. Materials and Methods

### 4.1. Chemical Mutagenesis and Seed Selection

A modified protocol of chemical mutagenesis was described previously [33]. We tested several concentrations of EMS ranging from 0.1% to 2% and selected the final 1% concentration based on the germination ability of the seed. At 1% concentration, the germination rate was approx. 70%, which is the recommended germination rate that ensures the best probability of random mutation in selected genes/pathways, whilst not decreasing the plant’s ability to grow.

Briefly, 20 g of seeds from the selected medium- to high-morphine-producing variety (1.11%) O-P-P19 were treated by 1% EMS for 60 min in aluminum-coated flasks. Prior to the EMS treatment, seeds were placed in water for 18 h to swell. After treatment, the seeds were placed into a sieve and washed with flowing tap water for 30 min. After washing, the seeds were planted into flowering pots in a common gardening substrate. From the chemically treated seeds, 195 plants were able to grow into mature plants and produce seeds in the M1 generation. The following year, 96 plant genotypes were selected randomly for the amplicon sequencing of three genes.

### 4.2. Mutation Screening

Seeds produced in the M1 generation were divided for mutation screening and for field cultivation. For mutation screening, 10 seeds of each genotype were planted in 5 cm diameter flowering pots and cultivated in growth chambers (Binder, Tuttlingen, Germany) in 16/8 light (150 μM photons/cm^2^)/dark period and watered daily with 5 mL of tap water. After 21 days of cultivation, a total of 50 mg of leaves from five plants were pooled and isolated by the EliGene Plant DNA Isolation Kit (Elisabeth Pharmacon, Brno, Czech Republic), according to the manufacturer’s instructions, and diluted to the same concentration of 5 ng/μL nanophotometer P300 (Implen, München, Germany).

For selected genes (see Section 2.1), primer sequences were selected by the NCBI Primer-BLAST [34] to span as large a part of the gene as possible, and optimized to melting temperature 60 ± 1 °C, self-complementarity < 5 and self-3′ complementarity < 2. Sequences for attaching barcoding primers were added to the primer sequences, and primers were produced with C6-amino-blocking modification (Elisabeth Pharmacon, Brno, Czech Republic). Primer sequences can be found in Table 4.

For the PCR reaction, EliZyme HiFi (Elisabeth Pharmacon, Brno, Czech Republic) polymerase was used, according to the manufacturer’s instructions, and 20 cycles were used. The specificity and effectivity of reactions were checked on 1% agarose gels. For the attachment of the barcoded primers, (SMRTbell 96 barcoded adapter plate 2.0; Pacific Biosciences, Menlo Park, CA, USA) 5 µL of previous PCR products was transferred onto new 96-well plates, mixed with 1 µL of barcoding primers, and the PCR reaction using EliZyme HiFi (Elisabeth Pharmacon, Brno, Czech Republic) polymerase was performed for another 20 cycles.

Quality control, library preparation and SEQUEL-SMRT sequencing were performed at the DNAlink company (Seoul, Republic of Korea) using their internal standard protocols.

Mutation screening was performed using Snippy 4.5.0 [35] on the Galaxy webserver [36] using standard parameters. Confirmation of found mutations was performed by searching for unique mutated sequences with an in-house written R script.

### 4.3. Field Cultivation

The seeds of the selected plants were sown with a Haldrup small-plot seeder in breeding nurseries at Opava- Kylešovice, Czech Republic (GPS: 49.9085183 N, 17.9354483 E). The date of sowing was 13 April 2022. The seed of each plant was sown in a plot of 2.5 m^2^ (1.25 × 2 m) in 5 rows and the distance between the rows was 25 cm. An amount of 0.25 g of seed per plot was sown (approx. 500 seeds). During the growing season, the plots were treated with herbicides and insecticides registered for poppy, and in the spring, the plots were fertilized with nitrogen at a dose of 20 kg/ha. After ripening, 30 capsules were taken from each plot on 1 August 2022. The capsules were subsequently deseeded and the obtained empty capsules were sent to the laboratory for an analysis of the alkaloid content. In parallel, the seeds of the guided mutant genotypes were sown in planters at OSEVA Development and Research Ltd., Research Institute of Oilseed Crops, at Opava (GPS: 49.9305269 N, 17.8783869 E). Three plants of each genotype were cultured. During flowering, cross-pollination of flowers was prevented using technical insulators (a paper bag placed over the flower). This resulted in the seed of isolated mutant plants remaining viable for further work.

### 4.4. RNA Preparation for Gene Expression Measurements

Sample collection and preparation for gene expression profiling were performed in three developmental stages. Whole buds, flowers or capsules were cut in the field, placed into ice and transferred into the laboratory where thin slices through the whole sample, or through the epidermis/phloem, were made. Approx. 100 mg of sample were homogenized in 1 mL of TRIzol (Molecular Research Center, Inc., Cincinnati, OH, USA) reagent and homogenized for 5000 rpm 90 s three times (MiniLys, Bertin, France). After each homogenization, samples were placed in ice for 90 s. Isolation was performed according to the manufacturer’s instructions. RNA concentration was measured using the NanoPhotometer P300 (Implen, Germany). Isolated RNA integrity was checked in randomly selected samples using bleach gel electrophoresis [37]. Reverse transcription was performed using EliZyme Reverse Transcriptase (Elisabeth Pharmacon, Brno-Židenice, Czech Republic), with a maximum recommended amount of whole RNA (400 ng) using a 1:1 ratio of oligo(dT) and random hexamer primers (Invitrogen, Waltham, MA, USA) as per the manufacturer’s instructions.

### 4.5. Gene Expression Profiling

A qPCR was performed using the EliGene AddRox reaction mix. The cDNA was then diluted 30 times, and reactions were performed in 10 µL in Eppendorf twin-tec 384-well plates. Each reaction consisted of 2 µL of cDNA, 2 µL of primer mix (0.5 µM of each primer), 5 µL of 2× master mix, and 1 µL of nuclease-free water. Each gene was measured in duplicate, in two technical replications using the 480 LightCycler (Roche, Basel, Switzerland). Relative expression fold-change was calculated using the ΔΔCq method [38], using actin as a reference gene and the original genotype as a control. Primer sequences were selected by NCBI Primer-BLAST [34] to span a central region of each gene, optimized to span exon-exon junctions, optimized to melting temperature 60 ± 1 °C, self-complementarity < 5, self-3′ complementarity < 2, and with amplicon sizes 110 ± 20 bp. Primer sequences can be found in the Appendix A.

### 4.6. HPLC Assessment of Main Alkaloids

Samples of poppy straw were ground on a laboratory grinder (IKA, Staufen, Germany) and sieved through a 0.5 mm sieve. For the sample preparation, SPE isolation of alkaloids and mobile phase preparation were used, containing: acetic acid 99.8% p.a., ammonia 25% p.a., hydrochloric acid 35% p.a., acetonitrile, and methanol for HPLC, glacial acetic acid 100% p.a. and trimethylamine 99.5% (Sigma-Aldrich, Taufkirchen, Germany). For alkaloid determination, standard solutions of alkaloid dissolved in methanol were used (1 mg·mL^−1^ morphine sulfide pentahydrate, codeine sulfate, papaverine hydrochloride, thebaine, and 0.125 mg·mL^−1^ noscapine). Following this, 50 mg of plant material was mixed with 5 mL of 5% acetic acid in a centrifuge tube, which was placed in an ultrasonic bath for 30 min and centrifuged for 10 min at 3900 rpm. The extract was purified by solid phase extraction on a vacuum manifold (Labicom, Olomouc, Czech Republic). CHROMABOND HR-XC 200 mg/3 mL SPE columns (Macherey-Nagel, Düren, Germany) were used for the SPE isolation of the alkaloids. For the HPLC of the main alkaloids content, Dionex UltiMate 3000 (Thermo Scientific, Waltham, MA, USA) on a reversed-phase column was used, using gradient elution and UV detection. Analyses were performed on an Ascentis Expres F5 (5 μm) 150 mm × 4.6 mm I.D. column (Supelco, Bellefonte, PA, USA); mobile phase A contained 5% acetonitrile, and mobile phase B contained acetonitrile: glacial acetic acid: triethylamine (97:2:1, *v*/*v*), flow rate 1 mL·min^−1^, column temperature 30 °C, injection volume 50 μL, wavelength 284 nm, analysis time 30 min and max. pressure 350 bar.

## 5. Conclusions

The reduction in the morphine alkaloid content is an important breeding goal and a prerequisite for growing opium poppies for food purposes. However, classical breeding methods, including classical mutagenesis, are not very effective. By using the TILLING method, in combination with NGS amplicon sequencing, we demonstrated the effectiveness of this approach, and we selected a set of poppy genotypes with significantly altered morphine content. Some genotypes showed not only reduced morphine content, but surprisingly also increased thebaine content. Therefore, the obtained opium poppy genotypes will be incorporated into two breeding programs: a poppy breeding program for the food industry, and a poppy breeding program for thebaine production utilizable in the pharmaceutical industry. Our results show that screening for mutations in a limited number of genes of the biosynthetic pathway of morphine alkaloids does not provide complete information on all the changes that have occurred as a result of mutagenesis. An example is genotype 9042, where in addition to the reduction in morphine content due to mutations in the genes studied, there was also an increase in the thebaine content. Our further study will focus on the whole genome sequencing of the mutation-affected genotypes so that all of the changes in the genes of the entire biosynthetic pathway will be defined. These findings can then shed light on the significance of mutations in individual genes and elucidate the results of the HPLC analyses. Our results may be another drop in the ocean of arguments toward targeted mutagenesis and targeted gene editing, instead of random chemical mutagenesis. This would, however, require changes in legislation, especially in the European Union.

## Figures and Tables

**Figure 1 plants-12-01077-f001:**
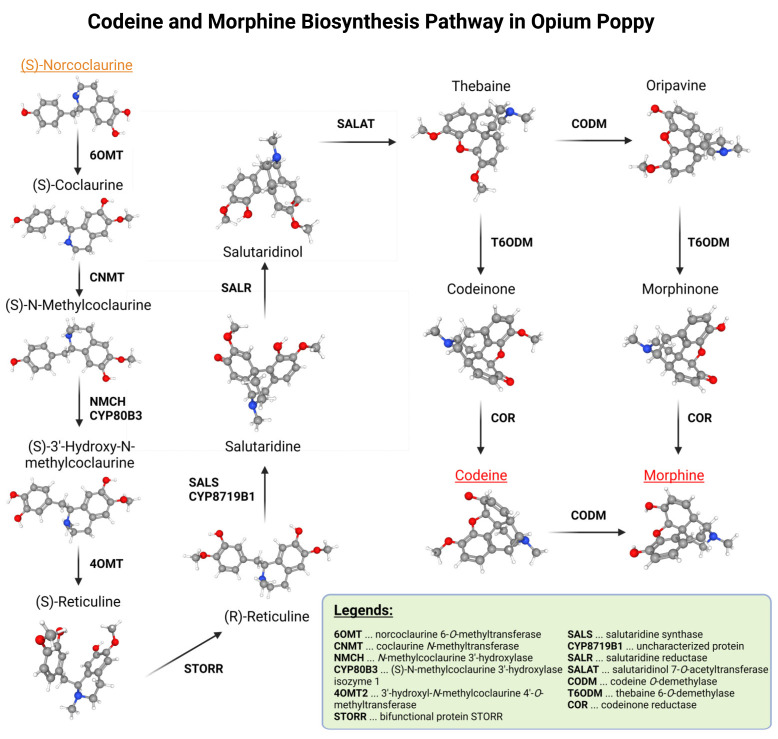
Morphine biosynthesis pathway without branches leading to different stable alkaloids.

**Figure 2 plants-12-01077-f002:**
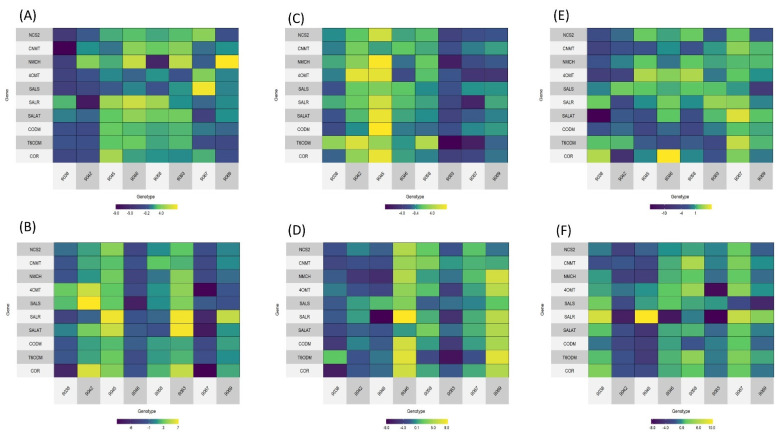
Heatmaps with gene expression profiles of mutant genotypes. Genes are sorted by their order in the morphine biosynthesis pathway. Blue color shows downregulation and yellow upregulation in comparison with the original variety. (**A**) Budding stage, epidermis. (**B**) Budding stage, slice. (**C**) Flowering stage, epidermis. (**D**) Flowering stage, slice. (**E**) Capsule stage, epidermis. (**F**) Capsule stage, slice.

**Table 1 plants-12-01077-t001:** Genes of morphine production pathway in opium poppy. The same genes are shown in the same color.

Gene ID	Gene Name	Shortcut	Gene Length	CDS Length	Chromosome	Location
113321914	salutaridine synthase	*SalS*	1959	1518	11	NC_039368.1 (128249007..128250966)
113339274	salutaridine synthase-like	*SalS*	1966	1521	Unplaced	NW_020631041.1 (8601134..8603100)
113322262	salutaridine reductase	*SalR*	1694	939	11	NC_039368.1 (128330274..128331968)
113340174	salutaridine reductase	*SalR*	1762	942	Unplaced	NW_020631041.1 (8574428..8576190)
113321357	salutaridinol 7-O-acetyltransferase	*SalAT*	2305	1425	11	NC_039368.1 (128319439..128321744)
113347789	thebaine 6-O-demethylase	*T6ODM*	2464	1095	2	NC_039359.1 (59955747..59958211)
113347787	thebaine 6-O-demethylase-like	*T6ODM*	2532	1095	2	NC_039359.1 (59933446..59935978)
113347785	thebaine 6-O-demethylase-like	*T6ODM*	2532	1095	2	NC_039359.1 (59911109..59913641)
113311621	codeine O-demethylase	*CODM*	1966	1083	1	NC_039358.1 (198606455..198608421)
113311630	codeine O-demethylase	*CODM*	1912	1083	1	NC_039358.1 (198677764..198679676)
113328201	NADPH-dependent codeinone reductase 1–5	*COR*	2000	966	Unplaced	NW_020619603.1 (7219550..7221550)
113294469	NADPH-dependent codeinone reductase 1–4	*COR*	1840	966	7	NC_039364.1 (2467198..2469038)
113340282	S-norcoclaurine synthase 2	*NCS2*	2912	2103	Unplaced	NW_020631041.1 (12957304..12960215)
113283898	(RS)-norcoclaurine 6-O-methyltransferase	*6OMT*	1355	1041	5	NC_039362.1 (183999440..184000794)
113294670	(S)-coclaurine N-methyltransferase	*CNMT*	2208	1056	7	NC_039364.1 (4136348..4138555)
113314340	(S)-N-methylcoclaurine 3′-hydroxylase	*NMCH*	1799	1464	1	NC_039358.1 (208684986..208686784)
113328451	(S)-N-methylcoclaurine 3′-hydroxylase	*NMCH*	2033	1467	Unplaced	NW_020619603.1 (9494942..9496974)
113283892	(S)-N-methylcoclaurine 3′-hydroxylase	*NMCH*	1802	1464	5	NC_039362.1 (183738532..183740333)
113339850	3’-hydroxy-N-methyl-(S)-coclaurine 4’-O-methyltransferase 2	*4OMT*	1493	1074	Unplaced	NW_020631041.1 (11170945..11172437)
113327792	3’-hydroxy-N-methyl-(S)-coclaurine 4’-O-methyltransferase 1	*4OMT*	1589	1065	Unplaced	NW_020619603.1 (9452525..9454113)

**Table 2 plants-12-01077-t002:** List of mutations found in M2 generation in selected genes. Nucleotide positions within CDSs are indicated, ‘->‘ stands for substitution, and ‘ins’ for insertion.

Mutation Found	Percent of Reads	Progeny Plants Name
*SalAT* 2011 30nt ins	90	9038–9043
*CNMT* 1205 A->G	6	9044–9045
*CNMT* 1211 T->A	6	9046–9049
*CNMT* 1256 A->T	7	9050–9055
*CNMT* 1456 A->T	6	9056–9060
*CNMT* 1695 G->A	5	9061–9070

**Table 3 plants-12-01077-t003:** Content of main alkaloids in selected M3 plants assessed by HPLC method. Numbers express the percent of the total sample volume. Genotypes selected for RT-qPCR analysis are marked in bold.

Sample/Plant Name	Morphine	Codeine	Thebaine	Sample/Plant Name	Morphine	Codeine	Thebaine
Control O-P-P19 original variety	1.453	0.110	0.296	9054	0.561	0.056	0.204
**9038**	**0.434**	**0.114**	**0.206**	9055	0.743	0.095	0.153
9039	0.551	0.070	0.210	9056	0.673	0.073	0.224
9040	0.507	0.079	0.380	9057	0.750	0.067	0.224
9041	0.636	0.094	0.304	**9058**	**0.200**	**0.117**	**0.145**
**9042**	**0.100**	**0.064**	**0.525**	9059	0.762	0.063	0.287
9043	0.635	0.096	0.235	9060	0.922	0.073	0.284
9044	0.972	0.077	0.300	9061	0.629	0.067	0.256
**9045**	**0.369**	**0.055**	**0.212**	9062	0.983	0.075	0.337
**9046**	**0.452**	**0.059**	**0.222**	**9063**	**0.571**	**0.055**	**0.199**
9047	0.868	0.085	0.278	9064	0.709	0.070	0.306
9048	0.527	0.104	0.304	9065	0.857	0.080	0.301
9049	0.471	0.101	0.210	9066	1.062	0.084	0.390
9050	0.485	0.076	0.324	**9067**	**0.378**	**0.094**	**0.166**
9051	0.763	0.088	0.336	9068	0.856	0.061	0.219
9052	0.513	0.064	0.224	**9069**	**0.615**	**0.044**	**0.202**
9053	0.892	0.091	0.338	9070	0.420	0.095	0.219

**Table 4 plants-12-01077-t004:** Primer sequences for mutation gene screening including barcoding overhangs. Left column gene name, forward (F) or reverse (R), and type of modification (C6-amino).

SALAT-F-C6	GCAGTCGAACATGTAGCTGACTCAGGTCACAAGAGGGGTATTCTATTCGGTGA
SALAT-R-C6	TGGATCACTTGTGCAAGCATCACATCGTAGGGATCATCTACCCGAAAACAACC
NCS2-F-C6	GCAGTCGAACATGTAGCTGACTCAGGTCACAAACACTCGGAACCGCAAGT
NCS2-R-C6	TGGATCACTTGTGCAAGCATCACATCGTAGGAAAAATCCTGAACCTTAGAGTGGA
CNMT-F-C6	GCAGTCGAACATGTAGCTGACTCAGGTCACGCTTGGGTTGATACCGGACC
CNMT-R-C6	TGGATCACTTGTGCAAGCATCACATCGTAGCGATGGTAAACAACACACAAAACG

## Data Availability

Due to specificity of data and intellectual property requirements by the funding agency, data will be provided upon request.

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
