# Peer review of "New Low Morphine Opium Poppy Genotype Obtained by TILLING Approach"

_plants, 2023, doi:10.3390/plants12051077_

Round 1

Reviewer 1 Report

The article "New low morphine Opium poppy genotype obtained by TILLING approach" deals with the comprehensive and systematic approach for getting a low morphine alkaloid in poppy using a combination of TILLING and NGS and validation of the mutants using RTPCR and HPLC etc. Only three of the single-copy genes of the morphine pathway among the eleven genes were used for the identification of mutants using TILLING and point mutations were obtained only in one gene CNMT  while an insertion in the other SalAT. Only a few expected transition SNP from G:C to A:T were obtained. There was no clear-cut explanation of the up or down-regulation of several genes of the alkaloids pathways investigated for getting low morphine mutants. The mutation for downregulation of genes upstream of the morphine synthesis must have occurred which has not been investigated using TILLING and NGS. It will be better to give the morphine biosynthetic pathway in the Materials and Methods. The article must be written in the past tense and in sound scientific English some of which have been highlighted with track changes in the revised manuscript attached. The reference section needs to be improved as per the journal format and all the scientific names of the plants should be italicized. In the conclusion section, it may be indicated that low morphine mutants of different genes of the pathway once identified should be pyramided for getting still low morphine poppy cultivar.

Author Response

The article "New low morphine Opium poppy genotype obtained by TILLING approach" deals with the comprehensive and systematic approach for getting a low morphine alkaloid in poppy using a combination of TILLING and NGS and validation of the mutants using RTPCR and HPLC etc. Only three of the single-copy genes of the morphine pathway among the eleven genes were used for the identification of mutants using TILLING and point mutations were obtained only in one gene CNMT  while an insertion in the other SalAT. Only a few expected transition SNP from G:C to A:T were obtained.

Thank you for the brief summarisation of the article.

 There was no clear-cut explanation of the up or down-regulation of several genes of the alkaloids pathways investigated for getting low morphine mutants. The mutation for downregulation of genes upstream of the morphine synthesis must have occurred which has not been investigated using TILLING and NGS.

Thank you for the comment. Unfortunately, we cannot be sure about that, but based on increased thebaine content (HPLC results) this might not be the case. In our opinion the mutation (that has also not been observed by NGS) is after the thebaine production step, as we have mentioned in the article. But this will be confirmed in future work.

 It will be better to give the morphine biosynthetic pathway in the Materials and Methods.

Thank you for the suggestion. We have prepared the pathway picture, and inserted into the manuscript, but if you agree, we have inserted into the results section, where it makes more sense.

 The article must be written in the past tense and in sound scientific English some of which have been highlighted with track changes in the revised manuscript attached.

Thank you for highlighting of the mistakes, we have taken care of them and of some more.

 The reference section needs to be improved as per the journal format and all the scientific names of the plants should be italicized.

Thank you for this comment, we have changed the reference style with the Zotero template, that have been provided by editors, and changed scientific names into italics.

In the conclusion section, it may be indicated that low morphine mutants of different genes of the pathway once identified should be pyramided for getting still low morphine poppy cultivar.

We have extended the discussion section and included this topic.

Reviewer 2 Report

The manuscript reported decreased production of morphine by a combination of TILLING approach with NGS sequencing. Further described the breeding process and characterization of alkaloid contents and gene expression profile for the main alkaloid-producing genes. The work is interesting and have potential to attract attention of wide readership from scientific community. However, the paper needs major revisions before acceptance, therefore before acceptance, ms must be revised considering following points/comments-

Abstract- I find this preliminary, please revised with a focus.

What was reason or basis to select only one mutagen and its concentration. Further for RT-PCR / gene expression profiling a few genes were selected, the basis of their selection need to be mentioned.  

As result mentioned that there is no pattern of gene expression and correlation of expression with HPCL data, then how authors interested the results and justification of selection of these gene with respect to targeted objective.  

How author justify the unexpected threats of obtaining unwanted results.

I strongly feel that discussion part can be improved.

Introduction part I find good.

Thanks / Good luck

Author Response

Abstract- I find this preliminary, please revised with a focus.

Thank you for this comment, as this is a short communication and not a whole article, we have tried to keep the abstract brief, but it was probably too much. We have rewritten and extended the abstract.

What was reason or basis to select only one mutagen and its concentration. Further for RT-PCR / gene expression profiling a few genes were selected, the basis of their selection need to be mentioned. 

Thank you for this comment, we have provided both explanations to appropriate parts of the manuscript lines 215-220 and 154-155, explanations also here:

Based on previous work on Flax (Linum usitatissimum L.) we chose EMS as the best chemical mutagen. We have tested several concentrations ranging from 0.1% to 2% and selected final 1% concentration, based on seed ability to germinate. At 1% concentration the germination rate was approx. 70%, which is the recommended germination rate, that ensures best probability of random mutation in selected genes/pathways and not decreasing plant ability to grow.

For qPCR we have focused on the main morphine pathway (we have added picture of the pathway) starting from NCS gene (as it was one of the mutation candidates) and not following expression of any branches leading to other stable metabolites (Papaverine, Noscapine and Sanguinarine).

As result mentioned that there is no pattern of gene expression and correlation of expression with HPCL data, then how authors interested the results and justification of selection of these gene with respect to targeted objective. 

Thank you for this question. It is difficult to be sure. One of the possible reasons is, that as this is only the M3 generation, there is still heterogeneity among the plants. We still think that it is better to focus on genes, that are only present in one copy, and focus more on phenotypical changes, reasons are in the discussion section, that has been extended. We have mentioned this in the manuscript (lines 97-101 and 184-188)

How author justify the unexpected threats of obtaining unwanted results.

This is to present more to discussion about genetically modified organisms. In EU legislation, our approach (creating random mutations and selecting mutants) novel genotypes are not considered as GMO. While targeted mutagenesis either using CRISPR or viruses results in novel genotypes to be considered GMO. We wanted to point out, that this is illogical due to the unknown mutations that occur. Also, targeted mutagenesis doesn’t require as much worktime, consumables etc. We have added explanation line in the conclusion section (line 340-341).

I strongly feel that discussion part can be improved.

We have extended and rewritten discussion, hope now it’s better in your point of view.

Introduction part I find good.

Thank you.

Round 2

Reviewer 2 Report

Accepted